# Remodeling the Tumor Myeloid Landscape to Enhance Antitumor Antibody Immunotherapies

**DOI:** 10.3390/cancers13194904

**Published:** 2021-09-29

**Authors:** Khiyam Hussain, Mark S. Cragg, Stephen A. Beers

**Affiliations:** Centre for Cancer Immunology, School of Cancer Sciences, Faculty of Medicine, University of Southampton, Tremona Road, Southampton SO16 6YD, UK; K.Hussain@soton.ac.uk (K.H.); msc@soton.ac.uk (M.S.C.)

**Keywords:** tumor-associated macrophages, resistance, antibody-dependent cellular phagocytosis, antibody immunotherapy, phagocytosis checkpoints

## Abstract

**Simple Summary:**

In addition to cancer cells themselves, tumors consist of several cell types that either function to suppress or promote tumor growth and disease progression. Macrophages are amongst the most abundant of these cell types and possess contrasting abilities to either enhance tumor growth and spread or to destroy cancer cells. Anti-cancer therapeutics such as Rituximab, Herceptin and Cetuximab, are cornerstones of current treatment for cancer patients. These therapeutics are antibodies that directly bind to cancer cells and aid macrophages in detecting and destroying these cells, through a process known as antibody dependent cellular phagocytosis (ADCP). Although the use of anti-cancer antibodies to treat large established tumors has improved survival rates, significant numbers of patients do not respond to this type of therapy. Thus, recently there has been an intense focus on designing new therapies which alter macrophages inside tumors, so that they can more effectively eliminate cancer cells through ADCP. Several molecules on the surface of macrophages can be targeted with antibodies to improve their ability to recognize and eliminate cancer cells. Here we review the most promising of these new targets, which offer the potential to circumvent resistance to therapy in cancer patients.

**Abstract:**

Among the diverse tumor resident immune cell types, tumor-associated macrophages (TAMs) are often the most abundant, possess an anti-inflammatory phenotype, orchestrate tumor immune evasion and are frequently associated with poor prognosis. However, TAMs can also be harnessed to destroy antibody-opsonized tumor cells through the process of antibody-dependent cellular phagocytosis (ADCP). Clinically important tumor-targeting monoclonal antibodies (mAb) such as Rituximab, Herceptin and Cetuximab, function, at least in part, by inducing macrophages to eliminate tumor cells via ADCP. For IgG mAb, this is mediated by antibody-binding activating Fc gamma receptors (FcγR), with resultant phagocytic activity impacted by the level of co-engagement with the single inhibitory FcγRIIb. Approaches to enhance ADCP in the tumor microenvironment include the repolarization of TAMs to proinflammatory phenotypes or the direct augmentation of ADCP by targeting so-called ‘phagocytosis checkpoints’. Here we review the most promising new strategies targeting the cell surface molecules present on TAMs, which include the inhibition of ‘don’t eat me signals’ or targeting immunostimulatory pathways with agonistic mAb and small molecules to augment tumor-targeting mAb immunotherapies and overcome therapeutic resistance.

## 1. Introduction

Solid tumors harbor cancer cells, non-transformed immune cells, and stromal cells, which promote tumor growth and metastasis. Within the tumor microenvironment (TME), this compartment comprises fibroblasts, endothelial cells, and several types of hematopoietic cells. The latter cell types are tumor-infiltrating cells of the immune system, and include T cells, natural killer (NK) cells and myeloid cells [1,2,3,4]. Tumor resident myeloid cells include neutrophils, eosinophils, dendritic cells, and importantly, macrophages. Resident macrophage populations within tumors are termed tumor-associated macrophages (TAMs) and can comprise up to half of the tumor mass [5,6,7,8]. TAMs were initially thought to be important in antitumor immunity due to their intrinsic phagocytic and cytotoxic properties; however, several studies have reported that in most large, established tumors, these functions of macrophages are suppressed and subverted to support tumor growth [9,10,11]. To address this, attempts have been made to re-establish and harness the antitumor functions of TAMs, particularly in the context of monoclonal antibody (mAb) immunotherapy [10]. 

Current Food and Drug Administration (FDA)-approved mAb therapies can be broadly classified into two types. Cancer or ‘direct targeting’ mAbs such as Rituximab, Herceptin and Cetuximab, exert their antitumor activity at least in part by binding the cancer cells themselves and mediating cell killing [12,13,14,15]. A second class of mAb, the so-called ‘immune checkpoint inhibitors’, such as Ipilimumab and Nivolumab, instead target and modify the anti-cancer effector functions of immune cells, most notably cytotoxic T lymphocytes [16,17]. Although both types of mAbs can be effective at suppressing or even eliminating large, established tumors, a significant fraction of patients fail to respond to treatment or develop resistance [18,19,20]. An improved understanding of the mechanisms underlying resistance to treatment is needed for the rational development of novel therapeutics that could augment the efficacy of existing tumor-targeting mAbs. 

In the human body, IgG-bound pathogens are cleared through a combination of neutralization, activation of the complement cascade, antibody-dependent cellular cytotoxicity (ADCC), antibody-dependent cellular phagocytosis (ADCP) and induction of inflammation [21]. These same mechanisms are also engaged by direct targeting mAbs to eliminate cancer cells. In most solid malignancies, TAMs are thought to be important effector cells recruited by direct targeting mAbs to deplete cancer cells via ADCP [22]. TAMs express several cell surface proteins that facilitate or suppress the phagocytic uptake of IgG-opsonized cell targets [23], with several factors in the TME known to alter their expression. These include cytokines, chemokines, growth factors, and metabolites, as well as Toll-like receptor (TLR) and stimulator of interferon genes (STING) signaling [24,25,26], which modulate macrophage activation status. It is well established that whereas proinflammatory cytokines such as TNF-α and IFN-γ preferentially augment Fcγ receptor (FcγR)-dependent TAM phagocytic function [27], anti-inflammatory cytokines such as Interleukin (IL)-4 and IL-13 are detrimental to FcγR-dependent uptake of cancer cells via ADCP [28]. Here, we explore the various ways in which mononuclear phagocytes and particularly TAMs may be manipulated to enhance the therapeutic efficacy of tumor-targeting mAbs. 

## 2. Tumor-Associated Macrophages

Macrophages are highly plastic cells that respond and adapt to the TME in which they are resident [29,30]. Macrophage functions range from organogenesis, the capture and elimination of pathogens, tissue homeostasis, wound healing and tumorigenesis [5,6]. In solid malignancies, TAM populations can impact tumors through a multitude of complex and often opposing mechanisms, including those impacting; cell death, immunoregulation, and angiogenesis, with the net result being either pro- or antitumor. However, recently a consensus has emerged whereby most TAMs in large tumors are thought to contribute to tumor progression by increasing cancer cell invasiveness, angiogenesis and immunosuppression [23]. 

TAMs originate from both bone-marrow-derived hematopoietic and non-hematopoietic lineages [31,32]. In early tumorigenesis, tissue resident macrophages accumulate within tumors and account for the majority of TAMs [33]. In the brain, tissue resident macrophages (known as microglia) arise from the yolk sac, and are distinct from hematopoietic precursors in the yolk sac or fetal liver, and proliferate within tissues throughout adulthood [11]. Furthermore, it has been reported that in murine gliomas, typically only 25% of TAMs originate from circulating monocytes, with the majority derived from tissue resident microglia [34,35]. In the liver, macrophages called Kupffer cells arise from both the yolk sac and embryonic hematopoietic stem cells [36]. In adulthood, the tissue microenvironment determines to what extent circulating blood monocytes replace these tissue resident macrophages [33]. As tumors increase in size and intratumoral vascular networks form, monocytes become the dominant source of TAMs [5,37,38]. The recruitment of TAMs to tumor sites is mediated by previously resident TAMs, cancer cells, and fibroblasts, secreting a range of chemokines including: chemokines (C-C motif) ligand (CCL)2, CCL5, CCL7, and chemokine (C-X3-C motif) ligand 1 (CX3CL1), as well as cytokines such as macrophage colony-stimulating factor (M-CSF), granulocyte-macrophage colony-stimulating factor (GM-CSF), and vascular endothelial growth factor (VEGF) [39,40,41,42]. Furthermore, classical monocytes (CD14^hi^FcγRIIIa^lo^ in humans and CD11b^hi^Ly6C^+^ in mice) are recruited as a tumor progresses and differentiate into TAMs, often in a CCL2-CCR2-dependent manner. Indeed, inhibition of CCR2 signaling blocks TAM recruitment and thus reduces TAM frequency, improving the survival of tumor-burdened mice in certain murine tumor models [5]. 

TAMs acquire immunosuppressive or immunostimulatory gene expression patterns in response to the dynamic and varied TME in large tumors. The expression patterns of these genes can be loosely categorized as pro- or antitumor in the context of disease prognosis and their potential impact on anti-cancer therapies (Figure 1). TAMs typically express myeloid surface markers such as CD68, CD163 (class A macrophage scavenger receptor), CD206 (mannose receptor, C type 1), macrophage galactose-type lectin (MGL), macrophage receptor with collagenous structure (MARCO), programmed cell death ligand 1 (PD-L1) and V-domain Ig suppressor of T cell activation (VISTA) [43,44,45,46]. In particular, CD68, CD163 and CD206 are extensively used to identify and quantify TAMs, in addition to their being used as prognostic markers for several tumor types [47].

## 3. TAM Activation States

The intrinsic heterogeneity of macrophages was historically stratified into two broad activation states: M1 (for proinflammatory or classically activated macrophages) and M2 (for anti-inflammatory or alternatively activated macrophages) [49,50,51]. Although it is often reported that TAMs more closely resemble M2 macrophages, the M1/M2 dichotomy is now thought to be too reductionist and these states are likely to be examples within a spectrum of activation states [52,53]. M1 macrophages are generated following stimulation with the interferon-γ (IFN-γ) alone or in concert with bacterial components, e.g., lipopolysaccharide (LPS) or pro-inflammatory cytokines such as tumor necrosis factor-α (TNF-α) [9]. Phenotypic regulation of M1-like macrophages is regulated via multiple transcription factors, such as IRF-1, STAT-1 and NF-κB [54,55,56]. These induce a pro-inflammatory phenotype in M1 macrophages, which is additionally regulated by the transcription factors: IRF-4, STAT6, PPAR-γ, the protein degradation adaptor protein, Tribbles homolog 1 (TRIB1), and chromatin modifiers including, histone demethylases and Jumonji domain-containing protein D3 (JMJD3) [55,57]. In contrast, M2 macrophages are polarized by several factors, and can be further subdivided into M2a, M2b, and M2c [9]. M2a macrophages are generated following exposure to IL-4 and/or IL-13. M2b macrophages are induced by immune complexes (ICs), LPS, certain Toll-like receptor (TLRs) agonists, or the IL-1 receptor antagonist (IL-1ra) [58,59,60]. M2c macrophages can be induced in response to exposure to IL-10, transforming growth factor-β (TGF-β), or glucocorticoids (GCs) [61,62]. TAMs with enhanced expression of CD163, CD204, CD206, stabilin-1, arginase-1, and matrix metallopeptidase 9 (MMP9), and elevated production of IL-10, VEGF, and prostaglandin E_2_ (PGE_2_), generally show M2-like characteristics [4,9,30,63]. In addition to implications for tumor neogenesis as well as “wound healing”, the status of ‘M2-like’ macrophages has ramifications for multiple treatment modalities.

In the TME, multiple factors polarize TAMs towards protumor anti-inflammatory phenotypes. Mediators released by tumor-infiltrating lymphocytes, such as T helper 2 (Th2)-cell-derived IL-4 and regulatory T (Treg) cell, and tumor-cell-derived IL-10, VEGF-A, TGF-β, and PGE_2_ activate an immunosuppressive program in TAMs [11,64]. Furthermore, in murine and human melanoma, IL-1β has been shown to induce the expression of Ten-Eleven-Translocation-2, a DNA methylcytosine dioxygenase, sustaining the immunosuppressive functions of TAMs [65]. Additionally, cancer-cell-derived CCL2, CCL18, CCL17 and CXCL4 work in concert to polarize macrophages towards M2-like phenotypes [66,67,68,69]. 

Hypoxia plays a pivotal role within tumors in regulating monocytes and macrophages, stimulating them to release factors that facilitate tumor growth, immunosuppression, and angiogenesis [70,71,72,73]. In the TME, hypoxia arises when cellular demand for molecular oxygen (O_2_) exceeds supply [74]. Hypoxia is a distinctive aspect of a wide range of solid tumors [75,76,77,78,79,80,81,82], and over half of tumor regions exhibit lower O_2_ levels than their healthy tissue counterparts [83]. For example, in pancreatic ductal adenocarcinoma, median *pO_2_* is 0–5.3 mmHg (0–0.7%) compared to 24.3–92.7 mmHg (3.2–12.3%) in healthy pancreata [78]. Under these conditions, lactate produced by tumor cells, as a by-product of aerobic or anaerobic glycolysis, stimulates TAMs to secrete elevated levels of Arg1 and VEGF [25]. 

## 4. Protumor Functions of TAMs

As described earlier, TAMs possess an ‘M2-like’ phenotype and function that promotes immunosuppression, metastases, and angiogenesis (Figure 2). Tissue resident macrophages and TAMs can phagocytose, and lyse cancer cells, activate NK cells and induce T helper 1 (Th1) immune responses [72,84,85]. However, TAMs are broadly associated with poor prognosis in several tumor types, including cholangiocarcinoma, glioma, Hodgkin lymphoma and ovarian and breast cancers [63]. Increased frequencies of CD163^+^, CD204^+^ and CD206^+^ TAMs correlate with tumor progression and worse clinical prognosis [54]. Furthermore, in some malignant tumors, the density and quantity of TAM infiltration is associated with higher Ki-67 expression, indicating elevated cancer cell proliferation [86]. 

TAMs also produce high levels of cytokines and chemokines, which recruit or induce immunosuppressive cell types at tumor sites. Thymically derived natural T_reg_ traffic and infiltrate to tumor sites via several chemokine receptors, in particular CCR4 [87]. Furthermore, abundant TAM production of CCL17, CCL18, CCL20 and CCL22 recruits CCR4^+^CCR6^+^ T_reg_ cells that actively suppress antitumor effector T cell responses [88]. Through the secretion of PGE_2_, IDO, IL-10 and TGF-β [9,61] TAMs promote the expression of the master Treg lineage transcription factor, Foxp3, as well as CTLA-4 in CD4^+^ T cells in the TME, inducing immunosuppressive Treg cell phenotypes. TAMs additionally recruit potently immunosuppressive myeloid-derived suppressor cells to tumor sites, which consist of immature monocytes and neutrophils [89]. 

In addition to inducing Treg cells and MDSCs at the tumor site, TAMs actively participate in the immunosuppression of effector T cells. TAM-derived arginine and tryptophan suppresses CD3 ζ-chain expression in T cells, resulting in the inhibition of effector T cell activation [90,91]. It has also been reported that when macrophages are cocultured with tumor cells under hypoxic conditions, they upregulate IDO, resulting in the suppression of T cell proliferation and IFN-γ production by effector T cells [92]. Furthermore, TAMs express the ligands of PD-1 (PD-L1, PD-L2) and CTLA-4 (B7 molecules), as well as additional checkpoints such as VISTA, which suppress effector T cell responses and promote Treg cell recruitment to the tumor site [10,93,94]. TAM-derived IL-10 also inhibits IL-12 expression in the TME, a cytokine essential for NK-cell cytotoxicity and the induction of Th1 responses [95].

TAMs express several enzymes, cytokines and chemokines that promote tumor metastases, such that TAM frequencies positively correlate with cancer cell invasiveness and metastasis [96]. Cancer-cell-derived M-CSF and TAM-derived EGF promote cancer cell migration along collagen fibers to aggregate around vasculature, and TAMs further release proteolytic enzymes such as MMPs, which destroy the extracellular matrix, promoting the dissemination of tumor cells [97,98]. CCL18 released by TAMs in MDA-MB-231 breast cancer tumors promotes the invasiveness of cancer cells by inducing integrin clustering, enhancing their adherence to the extracellular matrix, which is mediated by the CCL18 receptor PITPNM3 [99]. 

TAMs are also important promoters of angiogenesis in the TME. They function to degrade the tumor basement membrane, via the production of MMPs and cathepsins, and secrete proangiogenic growth factors such as VEGF, PDGF, bFGF and TGF-β that induce new vasculature in growing tumors [100,101]. Furthermore, TAMs are crucial promoters of the neoangiogenic switch in tumors. Hypoxia induces HIF-1α and HIF-2α expression in TAMs, which mediate hypoxia-responsive proangiogenic genes. Accordingly, TAM frequency correlates with the vascular density in murine and human tumors [102], and macrophage depletion in mice via clodronate treatment has been reported to suppress angiogenesis in several murine tumor models [103]. Angiopoietin 2 (ANGPT2) is a proangiogenic cytokine that is expressed by endothelial cells in tumors [104]. Expression of its receptor, TIE2, defines a highly proangiogenic subpopulation of myeloid cells, “TIE2-expressing monocytes/macrophages (TEMs)” [105]. Genetic depletion of TEMs markedly reduces tumor angiogenesis in various tumor models, emphasizing their role in tumor progression [106]. 

Cancer stem cells (CSCs) represent a subset of cells in the tumor that possess enhanced functions to promote tumor progression and metastasis [107]. TAMs interact with and enhance the tumorigenicity of CSCs via multiple mechanisms, which include the release of milk-fat globule-epidermal growth factor–VIII mediated by the STAT3 and sonic hedgehog pathways [108].

In summary, TAMs promote tumor growth through multiple mechanisms that are attributed to ‘M2-like’ phenotypes induced within the TME, highlighting a need to develop strategies that either delete these cells or repolarize them to proinflammatory antitumor states. In the context of direct targeting mAb immunotherapies, TAMs can function to phagocytose mAb-opsonized cells, and novel strategies to target the so-called ‘phagocytosis checkpoints’ to enhance the phagocytic functions of these cells are also currently under investigation.

## 5. TAM-Mediated Depletion of Cancer Cells

Tumor-targeting mAbs such as Rituximab, Herceptin and Cetuximab, recruit ADCP-mediating macrophages to directly eliminate cancer cells [109,110,111,112,113,114]. Checkpoint inhibitor mAbs such as Ipilimumab were previously thought to function solely via receptor blockade and expansion of effector T cells [115]. However, additionally, Ipilimumab has been reported to work optimally through the depletion of tumor-infiltrating immunosuppressive Treg cells, also indicating a role for ADCP-mediating myeloid cells [116,117,118]. Although several cell types are functionally capable of phagocytosing and destroying host cells, including epithelial cells, mesenchymal cells and fibroblasts, neutrophils, and monocytes [119,120,121], macrophages are ‘professional phagocytes’ and the principal effector cells in efferocytosis (clearance of apoptotic cells) and ADCP [122]. 

IgG antibodies can trigger ADCP indirectly via activation of the classical complement pathway, where iC3b-opsonized target cells can bind to complement receptor 3 (CR3, integrin α_M_β_2_) to elicit engulfment by ‘sinking phagocytosis’ [123]. Importantly, the macrophage cell surface receptors required for ADCP are less varied than for efferocytosis, with ADCP in the context of mAbs like Rituximab almost entirely dependent on FcγRs that bind the Fc portion of IgG antibodies. Human macrophages express the activating high affinity FcγRI and low affinity FcγRIIa and FcγRIIIa [21,124], as well as the inhibitory FcγRIIb. Antibody-bound target cells interact with FcγRI, FcγRIIa and FcγRIIIa for optimal ADCP (FcγRI, FcγRIII and FcγRIV in the mouse), whereas engagement with the sole inhibitory FcγR, FcγRIIb (FcγRII in mice), attenuates phagocytic function [21]. The expression levels and cellular distribution of FcγR on effector cells are of crucial importance in mAb therapy outcomes. Furthermore, human IgG1 and murine IgG2a, and IgG2c isotypes preferentially engage, activating above inhibitory FcγR, eliciting stronger ADCP (relative to human IgG2 or murine IgG1), and therefore are the preferred IgG isotypes for direct tumor-targeting mAbs [22,125,126]. 

After engagement, activating FcγRs cluster and phosphorylate ITAM in their cytoplasmic domains or associated gamma chains [21]. This induces the formation of the phagocytic synapse and thence, actin polymerization leads to the formation of the phagocytic cup [127]. The macrophage then extends pseudopodia around the opsonized target cell, engulfing it in a process termed zippering phagocytosis [128]. Actin filaments subsequently rearrange within the macrophage, causing its cell membrane to encompass the target cell, which leads to its inclusion into a phagosome. The phagosome fuses with endosomes and then lysosomes [129], followed by a marked reduction in pH (∼4.5) and generation of ROS [130], leading to the destruction of the phagocytosed cell [131]. The inhibitory FcγRIIb possesses an ITIM in its cytoplasmic domain, and the interaction of IgG Fc regions or immune complexes results in the recruitment of src homology 2 (SH2) domain containing inositol polyphosphate 5-phosphatase (SHIP), curtailing signaling from activating FcγR and consequently ADCP [132]. 

A seminal study by Clynes et al. [133] observed that nude mice deficient in the common gamma chain (FcRγ^−/−^/nu/nu mice), which consequently lack expression and signaling from the activating FcγRs, were unable to control human breast carcinoma BT474M1 growth in response to trastuzumab treatment. This implicated a role for activating-FcγR-bearing myeloid cells and NK cells in therapeutic outcomes. Importantly, mice deficient in the inhibitory FcγRIIb showed potent antibody-mediated target cell killing. The latter result not only demonstrated that FcγR-dependent mechanisms contribute substantially to the action of direct targeting mAbs, but implicated macrophages as key effectors cells in direct targeting mAb immunotherapy, given that NK cells do not express FcγRIIb in mice or humans [133]. 

Subsequent studies using intravital microscopy have reported that following anti-CD20 mAb therapy in murine models, Kupffer cells in the liver sinusoids, phagocytose circulating mAb-opsonized malignant B lymphoma cells [130,134,135], including in human CD20 transgenic mice [135]. Anti-CD20 mediated depletion of lymphoma cells in adoptive transfer models or the Eμ-Myc B cell lymphoma model has been shown to be dependent on activating FcγRs. Furthermore, the clodronate-mediated elimination of macrophages abrogated anti-CD20 therapy in this mouse model, further highlighting the indispensable role of macrophages in malignant B cell depletion [136]. 

Although the TME can establish an immunosuppressive transcriptional program in TAMs, which favors diminished ADCP, several strategies have been investigated to reverse this. Macrophages express high levels of Toll-like receptors (TLRs), that recognize pathogen-associated molecular patterns (PAMPs), which initiate inflammatory immune responses in response to infection. Accordingly, TLR ligands trigger the secretion of immunostimulatory cytokines. Similarly, the stimulator of the interferon genes (STING) pathway is capable of the rapid production of inflammatory type I IFNs [137]. As a result, the TLR3 agonist polyinosinic-polycytidylic acid (Poly I:C), and especially the STING agonists 2′2-, 2′3′-, 3′3′-cGAMPs and DMXAA, enhance cytokine release and activating FcγR expression on macrophages, augmenting ADCP by murine bone-marrow-derived macrophages, in vitro. However, only STING agonists could reverse the suppressive FcγR profile of TAMs induced in a murine model of B cell lymphoma, providing strong adjuvant effects alongside anti-CD20 mAb therapy [26]. STING-agonist-treated macrophages are more M1-like, and recently it was reported that M1 macrophages displayed enhanced ADCP relative to M2 macrophages of Raji, A431, and SKBR3 cells, in the presence of relevant direct targeting mAbs [138]. Furthermore, Resiquimod (R848), a TLR7/8 agonist, has been shown to re-polarize TAMs from M2 to M1 phenotypes, leading to enhanced ADCP in vitro and in mouse xenograft models [139]. Imiquimod, a TLR7 agonist, is the only TLR agonist currently approved by the FDA, and has been utilized to treat basal and squamous cell carcinoma patients [140]. Currently, poly I:C, R848, and NKTR-262 (TLR7/8 agonist) and Tilsotolimod (TLR9 agonists) are being evaluated in early-phase clinical trials, either as adjuvants for cancer vaccines or in combination with mAb therapies [141]. The TLR3 agonist Polyinosinic-polycytidylic acid-poly-l-lysine carboxymethylcellulose (poly-ICLC) is one of the most trialed drugs in its class, and numerous phase I/II trials are combining poly-ICLC with antitumor vaccines and checkpoint inhibitors in patients with advanced malignancies [142]. Finally, the TLR8 agonist Motolimod, in combination with Cetuximab, induces partial responses in metastatic head and neck cancer patients [143].

In addition to TLR agonists, several STING agonists, including BMS-986301 and GSK3745417, are in early phase trials as single agents or in combination with checkpoint inhibitors or standard chemotherapies for the treatment of a broad range of solid malignancies [144]. However, both TLR and STING agonists are also promising candidates for future combination strategies in the context of established direct targeting mAbs.

## 6. Antibody-Mediated Modulation of TAM Recruitment, Survival, and Effector Functions

Strategies to diminish the protumor functions of TAMs include the suppression of TAM generation, monocyte recruitment, and the repolarization of TAMs to proinflammatory phenotypes. Additionally, a compelling TAM targeting strategy has emerged that aims to target ‘phagocytosis checkpoints’ to enhance ‘eat me’ and block ‘don’t eat me’ signaling in tumors (Figure 3). Table 1 summarizes TAM-targeting mAbs in early phase trials that have been developed to reduce protumor TAM frequencies or augment antitumor immune responses in cancer patients. 

## 7. TAM Recruitment and Survival

### 7.1. CSF-1R

CSF-1R is a tyrosine kinase receptor expressed on all myeloid cells. Its ligands are M-CSF (CSF-1), GM-CSF (CSF-2) and IL-34, and their binding to CSF-1R induces differentiation, recruitment to tumor sites, and the survival of monocytes and macrophages [145]. The ‘M2-like’ TAM phenotype has been reported to be mediated by the growth factor M-CSF in addition to the Th2 cytokines: IL-4/IL-13, and Treg-cell-derived IL-10, in the TME [9]. Since the presence of CSF-1R^+^ TAMs correlates with poor survival in several tumor types [146], targeting CSF-1R represents an attractive strategy to eliminate or potentially repolarize these cells. Mononuclear phagocytes are almost completely absent in *CSFR1*^−/−^ mice [147]. Accordingly, mAbs targeting either CSF-1R or its ligand M-CSF have been developed. The antitumor and antimetastatic activities of anti-CSF-1R mAb have been demonstrated in subcutaneous EL4 lymphoma and MMTV-PyMT breast tumor models [148]. Clinical trials of the humanized anti-CSF-R1 mAb RG7155 (Emactuzumab, Roche) are currently underway in patients with solid malignancies, either as monotherapy or in combination with chemotherapy or mAb immunotherapy. It has been reported to reduce the number of TAMs and increase the CD8^+^/CD4^+^ T cell ratio in diffuse-type giant cell tumor (dt-GCT) patients (a macrophage-rich tumor type), consequently delaying tumor growth [149], with effects reported also in soft tissue sarcoma, mesothelioma, ovarian, breast and pancreatic cancer patients [150]. It was recently reported that chimeric antigen receptor (CAR) T-cell-mediated elimination of a subset of M2-like TAMs that express the folate receptor β (FRβ), led to enhanced tumor infiltration of pro-inflammatory monocytes and tumor-specific cytotoxic lymphocytes. This led to delayed tumor progression and enhanced survival of ID8-tumor-bearing C57BL/6 mice, highlighting the therapeutic potential of TAM depletion in the TME [151]. However, although CSF-1R blockade may eliminate immunosuppressive TAMs in the TME, it also has the potential to cause severe adverse events, such as opportunistic infections and diminished wound healing, since M-CSF and IL-34 are indispensable for maintaining normal macrophages for tissue homeostasis and pathogen elimination [152,153]. Furthermore, mononuclear phagocytes are the primary mediators of ADCP and so their elimination at tumor sites does not make anti-CSF-1R mAb blockade an attractive strategy to combine with established direct targeting mAbs. However, apart from decreasing TAM frequencies at tumor sites, targeting the CSF1/ CSF1R axis can also repolarize TAMs to an ‘M1-like’ phenotype. In a mouse proneural glioblastoma multiforme model, treatment with the small-molecule CSF1R inhibitor BLZ945 led to a reduction in M2-associated markers arginase 1 and CD206, but did not affect overall frequencies of TAMs [154].

### 7.2. CCR2/CCL2

CCL2 is a key chemokine which mediates macrophage recruitment to tumor sites. The anti-CCR2 mAb MLN1202 has been successfully used in patients at risk for atherosclerotic cardiovascular disease to reduce markers of inflammation [155]. Targeting CCR2 or its ligand, CCL2, with mAbs to block TAM recruitment has also been investigated in mice with orthotopic MDA-MB-231 human breast cancer tumors. Here, treatment with anti-CCL2 mAb reduced TAM accumulation, consequently reducing angiogenesis and tumor growth [156]. Furthermore, Carlumab, a human IgG1 anti-CCL2 mAb, has been investigated in clinical trials for patients with various solid tumors. However, this strategy was not sufficiently efficacious, even when combined with chemotherapy [157]. Likely explanations include the broad redundancy in the chemokine system, which contains dozens of different ligands and receptors. Indeed, tissue resident macrophages in particular, which differentiate into the most protumor fraction of the myeloid compartment, may be independent of regulation by any single chemokine receptor or ligand [158]. 

## 8. TAM Repolarization

### 8.1. CD40

CD40 is a compelling and widely investigated target for antitumor mAb immunotherapy. It is a TNF receptor superfamily member that is constitutively expressed on antigen-presenting cells (APCs) such as B cells and mononuclear phagocytes, as well as some non-haematopoietic cell types. Initial excitement around the targeting of CD40 with agonistic mAbs arose due to the generation of potent CD8^+^ T cell responses post-treatment in mouse models and patients; however, direct macrophage activation has now also become of significant interest [159,160]. In macrophages, CD40 stimulation leads to TNF receptor-associated factor (TRAF)-mediated intracellular signaling and cell activation [161]. In mononuclear phagocytes, it upregulates the expression of several costimulatory molecules, leading to enhanced antigen presentation and subsequent activation of cytotoxic T lymphocytes [162,163]. Myeloid cells, including CD11b^lo^F4/80^+^ macrophages, have been reported to enhance CD80, CD86 and MHC class II expression following treatment with anti-CD40 agonistic mAbs in mice bearing Pan02 tumors, improving overall survival [164]. The interaction of CD40 with its ligand (CD40L) or agonistic anti-CD40 mAbs also promotes the production of TNF-α, ROS and NO [165] in macrophages, which can all potentially enhance ADCP. Furthermore, in response to treatment with agonistic anti-CD40 mAbs in pre-clinical models, macrophages secrete IL-12, which is essential for the induction of the antitumor Th1 phenotype [166]. Using the LSL-Kras^G12D/+^;LSL-Trp53^R172H/+^;Pdx-1-Cre (KPC) model of pancreatic ductal adenocarcinoma, Beatty et al. observed that macrophages activated with an agonistic anti-CD40 mAb rapidly infiltrated tumors, participated in cancer cell killing, and facilitated the depletion of the tumor stroma. Importantly, these therapeutic effects were T-cell-independent, and re-education of TAMs alone in this model was sufficient to induce potent antitumor immunity [160]. Additionally, anti-CD40 mAb agonists induce intratumoral reorganization of the myeloid cell compartment by stimulating TAMs to induce MMP-dependent depletion of fibrosis [167]. These observations highlight the potential of macrophages in anti-CD40 mAb therapy. To date, several anti-CD40 mAbs have been investigated in clinical trials (Table 1), either as single agents or in combination with chemotherapy or checkpoint inhibitor therapy. Although these reagents have shown promise in inducing tumor regression or stabilizing disease, severe adverse events, including cytokine release syndrome and hepatotoxicity, have limited their clinical development [166]. Nonetheless, there has been a concerted interest in developing anti-CD40 mAb therapies that induce potent antitumor responses in patients. ChiLob7/4, APX005M, ADC-1013, and Dacetuzumab are humanized IgG1 mAbs. It has been reported that these mAbs depend on FcγR-mediated crosslinking to induce CD40 clustering and subsequent cellular activation. In contrast, Selicrelumab and CDX-1140 are humanized IgG2 mAbs, and potentially stimulate CD40 in an FcγR-independent manner [125,168,169]. This FcγR-independent biological activity is thought to be conferred through the conformational rigidity of the ‘B’ isoform of human IgG2, which more readily clusters CD40 at the cell surface, leading to enhanced intracellular signaling [170,171]. Currently, agonistic anti-CD40 mAbs are being combined with chemotherapy agents, TLR3 agonists, checkpoint inhibitors and Cabiralizumab (anti-CSF-1R). Furthermore, a clinical trial combining APX005M, Nivolumab and Ipilimumab is investigating its efficacy in melanoma and renal cell carcinoma patients [161]. Combining APX005M with Ipilimumab has the potential to induce enhanced ADCP-mediated depletion of immunosuppressive Treg cells in cancer patients [116]. 

### 8.2. PD-1

PD-1 is a co-inhibitory receptor most notably expressed on activated T cells. Its interactions with PD-L1 and PD-L2 expressed on both cancer cells and mononuclear phagocytes, results in downstream cell signaling that inhibits T cell ZAP70 phosphorylation, reducing its association with CD3ζ. This leads to the attenuation of intracellular signaling from the T-cell receptor and CD28, suppressing T cell activation [172]. PD-1 has been intensely investigated due to this potent inhibitory function on the immune system, and mAbs targeting PD-1 or PD-L1, such as nivolumab, pembrolizumab, and atezolizumab, have been developed and used as single agents or in combination with chemotherapy or other immunotherapies, to augment antitumor cytotoxic T cell responses in cancer patients [173,174,175]. Anti-PD-1/PD-L1 mAbs are amongst the most successful checkpoint inhibitors, having demonstrated considerable efficacy in the clinic and transformed the treatment of several previously incurable cancers [174]. Although the expression and function of PD-1 on T cells is well characterized, its expression on mononuclear phagocytes, particularly on macrophages and TAMs, has also been highlighted in recent years [44,176,177,178]. PD-1 expression on macrophages is associated with immunosuppression and polarization towards the ‘M2-like’ phenotype. PD-1 is upregulated on peritoneal macrophages in caecal ligation and puncture-mediated sepsis in mice, and in septic patients, in which it has been reported to be associated with immune dysfunction and an inability to control bacterial infection [176]. However, LPS and zymosan have also been reported to upregulate PD-1 expression in macrophages, through TLR4 and TLR2, respectively, which is mediated by NF-*κ*B activation, potentially regulating overzealous proinflammatory immune responses [178]. PD-1 has also been reported to regulate cytokine production in macrophages. Mononuclear phagocytes from hepatitis-C-infected patients express high levels of PD-1 and decreased expression of IL-12. Importantly, treatment with anti-PD1 restores IL-12 production and induces STAT1 phosphorylation in ex vivo monocytes from these patients [179]. In the context of cancer, using a human osteosarcoma pulmonary metastasis mouse model, Dhupakar et al. [177] demonstrated that anti-PD-1 therapy induces metastases regression by activating M1 TAMs and reducing M2 TAM frequencies [177]. A study by Gordon et al. [44] reported that TAMs, both in the murine CT26 subcutaneous tumor model and human colorectal tumors, expressed elevated levels of PD-1 relative to splenic mononuclear phagocytes or healthy peripheral blood. PD-1 expression was also shown to be higher on M2 TAMs relative to M1 TAMs in human colorectal tumors and was positively correlated with the expression of CD206 and the decreased expression of MHC class II. Importantly, these PD-1^+^ TAMs had a decreased ability to phagocytose fluorescently labelled CT26 cancer cells in vivo, a function that was restored after anti-PD-1 mAb treatment. [44]. These observations indicate that PD-1/PD-L1 targeting mAbs, in addition to augmenting antitumor cytotoxic T lymphocyte responses, also have the potential to repolarize TAMs to therapeutically beneficial proinflammatory phenotypes with restored phagocytic function. Clinical use of checkpoint inhibitors such as Nivolumab, in combination with direct targeting mAbs, therefore has the potential to augment the efficacy of direct targeting mAb therapy.

### 8.3. Macrophage Receptor with Collagenous Structure (MARCO)

MARCO is expressed on a subset of macrophages and mediates the sensing and elimination of pathogens through the recognition of PAMPs. It recognizes ligands that include nucleic acids, LPS, oxidized lipoproteins, and several endogenous proteins [180]. Recently, it was reported that MARCO^+^ TAMs suppress cytotoxic T lymphocyte and NK-cell activation, and conversely enhance Treg proliferation and IL-10 production. Targeting MARCO with mAbs (alongside CRISPR-mediated deletion of IL-37 receptor) in lung cancer cell lines repolarized TAMs, enhancing antitumoral killing capacity and immunostimulatory functions [181]. Furthermore, in pre-clinical models of breast cancer, colon cancer and melanoma, anti-MARCO mAbs have been reported to induce antitumor immunity by potently repolarizing TAMs to a proinflammatory phenotype [182]. Recently, it was reported that anti-MARCO mAbs suppressed angiogenesis, switched the metabolic program of MARCO^+^ TAMs, induced NK-cell killing through TNF-related apoptosis-inducing ligand (TRAIL) and synergized with anti-PD-1/PD-L1 mAb therapy, to enhance tumor suppression [183]. These promising observations indicate that antagonist anti-MARCO mAbs have the potential to repolarize TAMs to a proinflammatory phenotype and augment direct targeting mAb therapies.

### 8.4. V-domain Ig Suppressor of T Cell Activation (VISTA)

VISTA was identified in mice as an Ig superfamily inhibitory cell surface molecule that interacts with VSIG-3 and tumor-derived Galectin-9. It is predominantly expressed on hematopoietic cells and is abundantly expressed on CD11b^high^ blood monocytes [184,185]. Post-ipilimumab therapy in prostate cancer patients, VISTA is upregulated on CD68^+^ macrophages [186]. TAMs have also been reported to express VISTA in human colorectal carcinoma, and its expression on M2 macrophages is thought to contribute to the anti-inflammatory, pro-tumorigenic functions in these patients [187]. Indeed, agonistic anti-VISTA mAb treatment in vitro has been shown to prevent M1 polarization [188]. In vitro experimental data also show that mAb-mediated VISTA blockade on human monocytes augments their ability to activate T cells from prostate cancer patients [185]. These observations indicate that antagonistic anti-VISTA mAbs have the potential to repolarize TAMs to an M1 phenotype, providing evidence that they may augment other immunotherapies. 

### 8.5. Triggering-Receptor-Expressed on Myeloid Cells 2 (TREM2)

TREM2 is a member of the Ig-superfamily that transmits ligand-mediated inhibitory intracellular signaling. It is widely reported to be expressed on microglia in the brain, where it maintains metabolic homeostasis during physiological stress by binding to lipids, lipoproteins and amyloid-β, which are implicated in Alzheimer’s disease [189]. In addition to microglia, the expression of TREM2 has also been observed in several other types of tissue-resident macrophages [189]. Katzenelenbogen et al. [190] have recently reported that Arg1^+^TREM2^+^ TAMs and monocytic cells represent a key regulatory myeloid cell subset. Furthermore, *TREM2*^−/−^ mice were significantly protected from MCA205 tumor progression, which was associated with higher infiltration of cytotoxic T lymphocytes and NK cells. These observations support the view that TREM2 is a marker for immunosuppressive TAMs and monocytes [190]. Concurrent work in murine tumor models by Molgora et al. [191] demonstrated that treatment with anti-TREM2 mAb suppressed tumor growth, augmented antitumor effector T cell responses and reduced MRC1^+^CX_3_CR1^+^ macrophages in the tumor infiltrate when combined with anti-PD-1 mAb therapy [191]. As detailed above, TREM2 is known to recognize a wide range of ligands, but it remains unknown which specific ligands it recognizes in the TME [48]. Nevertheless, these recent studies indicate that antagonistic anti-TREM2 mAbs have the potential to augment antitumor T cells responses, although the impact of blockade in the context of direct targeting mAb immunotherapy and ADCP warrants further investigation.

### 8.6. CD204

CD204, also known as Scavenger receptor-A (SR-A), is expressed primarily on macrophages and dendritic cells [192], and is abundantly expressed on TAMs in several major tumor types, where it is a poor prognostic marker [193]. It is able to bind a broad range of ligands, including lipoproteins, LPS and several proteins expressed by apoptotic cells [194]. In atherosclerosis, CD204 has been identified as a major receptor on macrophages that mediate the uptake of oxidative or acylated low-density lipoproteins [195,196]. Additionally, CD204 has been reported to regulate the expression of cytokines and chemokines, primarily via the regulation of TLR- and cytokine-mediated cell activation [192,197]. Furthermore, loss of CD204 in mice causes impairment of host defense against early phase bacterial infections [198,199]. In addition, CD204^−/−^ mice exhibit an increased susceptibility to endotoxic shock [200], due to decreased clearance of LPS in the absence of CD204, leading to greater TLR4 signaling and inflammation [201]. In the context of cancer, it has been reported that heat shock protein-mediated antitumor activities and antitumor efficacy of vaccines using TLR agonists as adjuvants are enhanced in CD204^−/−^ mice [202]. The growth of EL4 lymphoma cells is delayed in CD204^−/−^ mice, although TAM frequencies within these tumors have been found to be comparable to tumors in wild type mice. In response to engulfment of necrotic lymphoma cells, CD204^−/−^ macrophages express enhanced levels of NO, IFN-β and IFN-γ, suggesting an important role of CD204 in regulating TAM function by inhibiting TLR and IFN signaling pathways [203]. These latter observations suggest that although CD204 mediates the phagocytosis of apoptotic cells, its inhibitory functions on cytokine signaling indicate that mAb-mediated CD204 antagonism may possess more utility in oncological settings. 

### 8.7. Leucocyte Immunoglobulin-Like Receptor B 2 (LILRB2)

LILRB2 is one of the best-characterized members of the inhibitory human LILRB family, and it binds to classical and non-classical HLA class I ligands [204,205], as well as to members of the angiopoietin-like protein family [206,207]. LILRB2 expression is largely restricted to myeloid cells, making it an attractive target for modifying the tumor myeloid cell landscape. LILRB2 binds HLA class I molecules at two binding sites, interacting with the a3 domain of the HLA-class I heavy chain and separately with β2M [208]. Inhibitory signaling by LILRB2 is mediated by its cytoplasmic ITIMs, which recruit the phosphatases SHP1 and SHP2 [209]. Several studies have shown that HLA-G:LILRB1/2 interactions increase IL-4 and IL-13, suppressing proinflammatory cytokine release and promoting production of IL-10 and TGF-β [210], which drives TAMs towards the ‘M2-like’ phenotype. In addition, LILRB2 has been reported to compete with CD8 for HLA class I binding, thereby potentially modulating cytotoxic T lymphocyte responses [207]. In vitro studies have revealed that the phagocytosis of several cancer lines is negatively correlated with HLA class I expression. Furthermore, HLA class I deficient B6-F10 tumor growth has been reported to be suppressed in immunodeficient non-obese diabetic (NOD)-*scid* IL2Rgamma^null^ mice, although LILRB2 gene deficiency did not enhance CD47 disruption-mediated cancer cell phagocytosis in vitro [211]. However, anti-LILRB2 mAb treatment has been reported to enhance proinflammatory gene expression in LPS-treated macrophages in vitro and in a mouse model of lung cancer. Anti-LILRB2 mAb treatment also enhances the response to anti-PD-1 mAb therapy and skews TAMs toward an immunostimulatory phenotype [212]. Thus, LILRB2 mAb-mediated blockade demonstrates a capacity for TAM reprogramming, in addition to potentially suppressing ‘don’t eat me’ signaling derived from interactions with HLA class I on cancer cells. 

### 8.8. Tyro3, Axl, and MerTK (TAM) Receptors

The TAM receptors Tyro3, Axl, and MerTK are expressed on several cell types, including macrophages, where they mediate polarization and efferocytosis [213]. The TAM receptors are receptor tyrosine kinases which share a similar conformational structure, and bind to two common ligands: protein S and Gas6 [214]. Phosphatidylserine (PtdSer) has been reported to strengthen the binding of TAM receptors to their ligands, enhancing intracellular signaling [215]. All three TAM receptors facilitate efferocytosis by macrophages; however, MerTK has been reported to be essential to the process, since the clearance of apoptotic thymocytes in *Mertk*^−/−^ mice is impaired [216]. Furthermore, phagocytic function is reduced in RAW264.7 macrophages treated with an anti-MerTK mAb [217]. It has also been suggested that although MerTK primarily mediates efferocytosis in the TME, Axl is more crucial to the uptake of apoptotic cells in inflammatory settings, such as during infection [215]. Both Axl and MerTK expression is higher on M2-like macrophages relative to M1-like macrophages [218]. Furthermore, Gas6-mediated MerTK triggering in RAW264.7 murine macrophages leads to M2-associated gene expression [219]. Due to this ability of TAM receptors to induce M2 polarization in macrophages, small molecule inhibitors, mAb-drug conjugates, Axl-Fc fusion proteins and CAR-T therapies targeting TAM receptors are currently in clinical trials [213,218]. It may be possible to develop an antagonistic pan-TAM mAb as a strategy to inhibit M2 polarization in the TME, due to receptor homology between the three TAM receptors. However, the broad expression of TAM receptors as well as preclinical evidence indicates that such a therapeutic would additionally suppress efferocytosis, and so would warrant further investigation in the context of the mononuclear phagocyte compartment in the TME. 

## 9. Phagocytosis Checkpoints

### 9.1. Activating FcγRs

As detailed above, FcγRs are the master regulators of ADCP, and their importance for direct targeting mAbs has been conclusively demonstrated by the observations that antitumor therapy is diminished in activating FcγR-deficient (γ chain^−/−^) mice, and conversely enhanced in FcγRIIb^−/−^ mice [133]. Targeting activating FcγR with agonistic mAbs that can augment the activation and phagocytic function of macrophages, without perturbing direct targeting mAb Fc:FcγR interactions, can potentially be developed to enhance therapeutic efficacy. However, due to adverse events, such as cytokine release syndromes, associated with the clinical use of mAbs targeting activating FcγR, these strategies were largely unsuccessful in clinical trials conducted in the 1990s. Early phase trials of the bispecific antibodies MDX-447 (humanized Fab’_2_ anti-FcγRI x humanized Fab’_2_ anti- EGFR), MDX-H210 (humanized Fab’_2_ anti-FcγRI x Fab’_2_ anti-HER2/neu) and MDX-33, an anti-FcγRI mAb, to treat several types of solid malignancies, resulted in monocytopenia and elevated serum cytokine levels. However, reductions in the size of metastatic lesions were observed in one RCC patient [220]. Although activating FcγRIIa is expressed on monocytes, macrophages, DCs and neutrophils, its abundant expression on platelets is a major obstacle to the clinical development of anti-FcγRIIa mAbs. Human FcγRIIa-expressing murine platelets are directly activated by IgG immune complexes in vivo, releasing proinflammatory cytokines, and are sufficient to restore susceptibility to anaphylaxis in resistant mice [221]. The pan anti-human FcγRII mAbs, IV.3, AT-10 and MDE-8 induce anaphylaxis in FcγRIIa transgenic mice. However, variants of these mAbs, as well as a recently developed anti-FcγRIIa antibody (VIB9600) lacking the capacity to engage FcγR via their Fc regions, failed to induce anaphylaxis or immune thrombocytopenic purpura (ITP), and protected FcγRIIa transgenic mice from near lethal doses of IgG ICs [222,223]. Nonetheless, the use of agonistic anti-FcγRIIa mAb in combination with direct targeting mAbs to enhance cancer cell elimination remains a challenge, due to the expression of FcγRIIa on platelets [221]. Clinical trials investigating 3G8, a murine IgG2a anti-human FcγRIIIa mAb in ITP patients, have also reported adverse events and elevated serum cytokine levels [224,225,226]. The response to 3G8 in a patient with refractory human immunodeficiency virus (HIV) resulted in significant NK cell activation and increased serum levels of TNF-α, IFN-γ and GM-SCF [224]. Nonetheless, the humanized GMA161 mAb was developed from 3G8; however, early phase trials in ITP patients caused severe adverse events that were not associated with cytokine or histamine release. However, treatment with an inhibitor of platelet-activating factor (PAF) eliminated all signs of hypersensitivity to GMA161 in these patients [226]. As more refined bispecific modalities develop, harnessing these powerful receptors for anti-cancer activity may become more amenable, but currently it remains difficult to separate activity from toxicity. 

### 9.2. FcγRIIb 

A potentially superior approach to modulating FcγRs to augment antitumor mAb therapy is to combine the use of these reagents with mAb against the inhibitory FcγRIIb. FcγRIIb is a key phagocytosis checkpoint molecule in the context of direct targeting mAb immunotherapy [116,227]. In addition to its inhibitory effect on mononuclear-phagocyte-mediated phagocytosis, FcγRIIb has been reported to limit the efficacy of anti-CD20 mAb therapy and to promote antibody drug resistance by additional mechanisms, when expressed on malignant B cells [228,229,230]. We have observed that FcγRIIb expressed on malignant B cells promotes the internalization of rituximab bound to CD20 on the surface of the same malignant B cells, decreasing CD20 availability for rituximab engagement, and consequently ADCP and ADCC. This cis interaction between the Fc portion of anti-CD20 mAb and FcγRIIb on the surface of the same malignant B cells results in mAb degradation, through a process termed antibody bipolar bridging [112]. Collectively, these observations provided a strong rationale to develop antagonistic anti-FcγRIIb antibodies that block FcγRIIb-mediated antibody internalization in order to augment macrophage-mediated ADCP, in combination with directing targeting mAbs [231]. Highly specific anti-human FcγRIIb has been generated despite ~93% homology between the extracellular domains of the ancestrally related activating FcγRIIa and inhibitory FcγRIIb [228]. The human IgG1 antagonistic anti-FcγRIIb antibody (6G11/BI-1206) synergizes with and augments rituximab-mediated B cell depletion in human CD20 transgenic mice, enhances anti-CD20 mAb-mediated ADCP, and diminishes the refractoriness of primary human chronic lymphoblastic cells to anti-CD20 mAb therapy in vivo in mice [228]. Currently, two clinical trials are underway to evaluate the safety and therapeutic efficacy of the anti-FcγRIIb mAb: BI-1206 antibody as a single agent and in combination with rituximab in patients with B cell malignancies (NCT03571568 and NCT02933320). Recent reports indicate that this approach is promising, with several previously rituximab-refractory mantle cell lymphoma patients responding to therapy (https://doi.org/10.1182/blood-2020-140219, accessed on 28 September 2021). 

### 9.3. CD47

CD47 is a ubiquitously expressed immunoglobulin superfamily member. It is expressed on red blood cells, as well as on all nucleated cells, and is importantly upregulated on some cancer cell types [232]. CD47 is the best known “don’t eat me” signal, and mediates resistance to cancer cell phagocytosis by macrophages in efferocytosis and ADCP. The ligand of CD47, SIRPα, is expressed on macrophages and TAMs [233]. In mononuclear phagocytes, CD47 binding to SIRPα leads to intracellular signaling and activation of SHP-1/SHP-2, which inhibits Rac activation downstream of phagocytic receptor stimulation [234]. Rac plays a crucial role in driving actin polymerization and target cell internalization during efferocytosis and FcγR-dependent phagocytosis [235]. Blockade of CD47 has been shown to restore phagocytosis and the clearance of tumor cells, and to induce tumor regression in several preclinical cancer models. The use of anti-CD47 mAb to control the growth of human tumor xenografts in NOD SCID gamma^TM^ mice was therapeutically efficacious for medulloblastoma, glioblastoma, ovarian, bladder, colorectal and breast tumors [236,237,238]. A treatment combining anti-CD47 mAb with TTI-621, a SIRPα-Fc fusion protein used to prevent SIRPα:CD47 interaction, was reported to promote the phagocytosis of tumor cells in a B-cell lymphoma mouse model [239,240]. Furthermore, non-functional SIRPα variant soluble proteins in preclinical models and anti-CD47 mAb in non-Hodgkin’s lymphoma patients have been combined with rituximab, resulting in tumor regression [241,242]. Advani et al. [241] reported that the CD47-blocking antibody Hu5F9-G4 had therapeutic efficacy in combination with rituximab in patients with diffuse large B-cell lymphoma and follicular lymphoma, and the authors proposed that blocking CD47–SIRPα interactions improved tumor cell phagocytosis by macrophages [241]. However, anti-CD47 mAb therapies may increase the occurrence of transient anaemia, since red blood cells also express CD47 [232]. Furthermore, in pre-clinical xenograft immunodeficient mouse models, anti-CD47 mAbs are used at high doses to induce tumor suppression, potentially over-estimating the likely efficacy of CD47 blockade in humans even in combination with direct targeting mAbs. To circumvent these obstacles, several groups, including Dheilly et al. [243] have developed bispecific antibodies that target CD47 with lower affinity alongside other tumor specific antigens, to achieve tumor-specific CD47 blockade, which spares red blood cells [243]. 

### 9.4. CD24

CD24 is a heavily glycosylated GPI-anchored surface protein [244] expressed on cancer cells, particularly in ovarian cancer and triple negative breast cancer, relative to healthy tissue [245]. Its ligand, Siglec-10, is expressed on B cells, dendritic cells, and macrophages, and has recently been reported to be abundantly expressed on TAMs in hepatocellular carcinoma (HCC) [246]. Siglec-10 contains two ITIMs in its cytoplasmic domain, and CD24/Siglec-10 interactions induce inhibitory signaling mediated by SHP-1/SHP-2, leading to the suppression of TLR-mediated inflammation and macrophage phagocytic function [247,248,249]. CD24 gene deletion in tumor cell lines or mAb-mediated blockade has been reported to enhance the phagocytosis of MCF-7 cancer cells in vitro, and MCF-7 tumor growth has been suppressed in vivo in CD24^−/−^ mice [245]. A clinical trial to test the safety and efficacy of a CD24-Fc fusion protein in combination with anti-CTLA4 and anti-PD-1 in metastatic melanoma, colon cancer, and renal cell carcinoma is currently in early phase trials (NCT04060407). The immunosuppressive function of Siglec-10 in macrophages and its restrictive cellular expression profile also make it an appealing target for mAb-mediated blockade. Abundant infiltration of Siglec-10^hi^ TAMs has been reported to be associated with diminished cytotoxic T cell immune responses in HCC, and whole transcriptome analysis has revealed marked ‘M2-like’ gene expression in Siglec-10^hi^ TAMs. Furthermore, Siglec-10 mAb-mediated blockade enhanced cytotoxic T-lymphocyte-mediated killing of HCC cells, and this function synergized with concurrent pembrolizumab treatment in vitro [246]. However, full evaluation of the therapeutic efficacy of mAb-mediated blockade of CD24:Siglec-10 interactions in cancer patients warrants further investigation.

## 10. Conclusions

Although TAMs are indispensable effector cells in direct targeting mAb immunotherapy, the immunosuppressive TME markedly reduces their ability to elicit ADCP and deplete mAb-opsonized targets. Currently, several early phase clinical trials are investigating different TAM-targeting mAbs, and in particular, anti-CD40 agonistic mAbs hold great potential to repolarize TAM activation states in the TME. Furthermore, mAb-mediated FcγRIIb blockade is also a promising candidate for the enhancement of ADCP in the context of anti-CD20 mAb therapies. Recently, high-dimensional single-cell RNA sequencing has shed new light on the variety of myeloid cells in the TME. Furthermore, this has also revealed novel TAM-associated cell surface markers and signaling pathways, with the potential for targeted intervention to reshape the tumor myeloid cell landscape to in turn enhance clinical outcomes. 

## Figures and Tables

**Figure 1 cancers-13-04904-f001:**
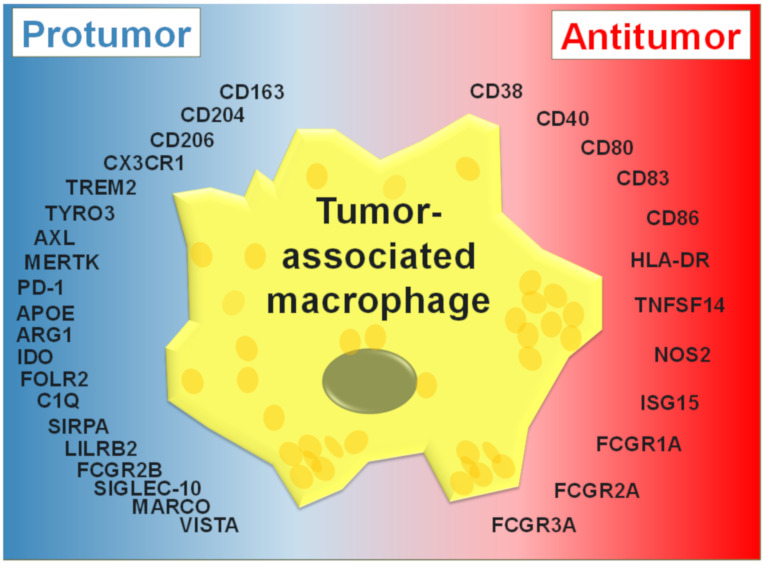
TAM-associated markers. Expression of genes in TAMs that phenotypically and functionally associate with protumor (blue) and antitumor (red) outcomes in the context of tumor progression and/or efficacy of direct targeting mAb therapy (adapted from [48]).

**Figure 2 cancers-13-04904-f002:**
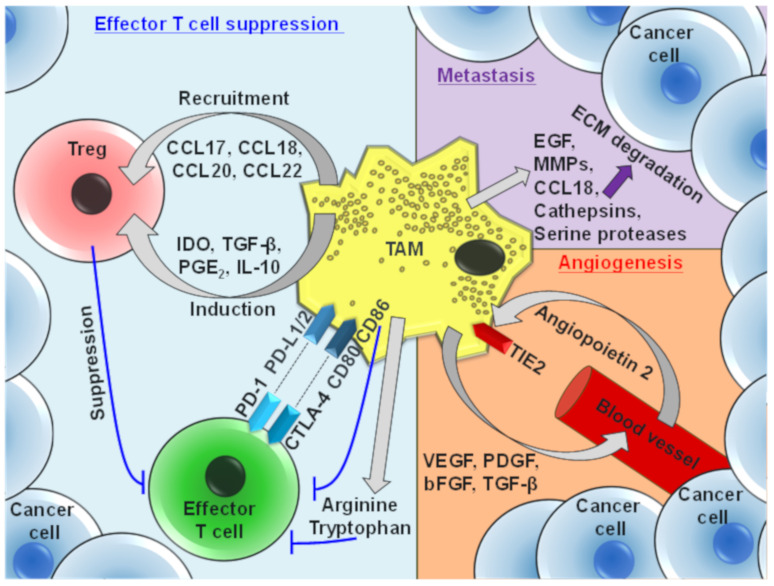
Major protumor functions of tumor-associated macrophages. TAMs mediate suppression of effector T cells via the secretion of soluble proteins and through the expression of inhibitory cell surface molecules. TAMs also produce several factors that promote extracellular matrix (ECM) degradation, which facilitates tumor metastasis. Furthermore, TAMs secrete cytokines that promote angiogenesis, consequently accelerating tumor growth.

**Figure 3 cancers-13-04904-f003:**
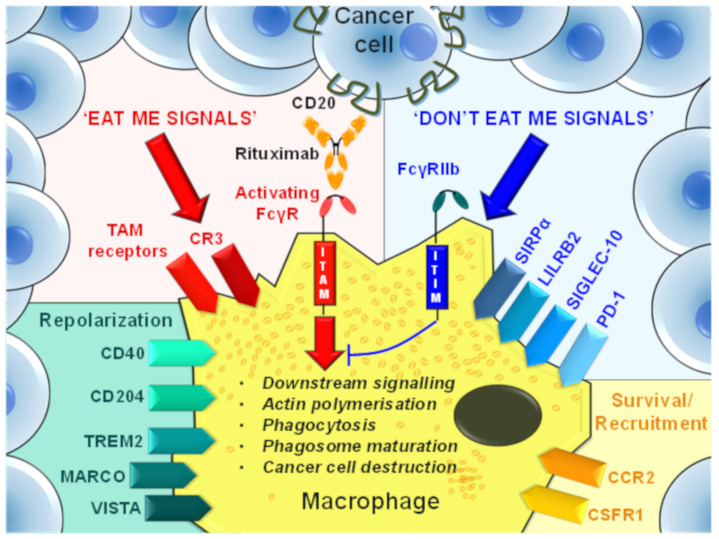
TAM cell surface molecule candidates for mAb targeting. TAM cell surface molecules that can potentially be targeted by mAbs to modify TAM frequencies or repolarization to a proinflammatory phenotype in the TME. These mAb targets are grouped according to the predominant effect resulting from stimulation of their natural ligand. However, mAb-mediated targeting of these molecules may exert further functional changes in TAMs and healthy mononuclear phagocytes.

**Table 1 cancers-13-04904-t001:** TAM-targeting mAbs in completed or active trials. These mAbs have been investigated in or are in active clinical trials, either as single agents, or in combination with chemotherapeutic agents, checkpoint inhibitors, Fc fusion proteins or TLR agonists.

Target	Compound	Sponsor	Phase	Indication	Status	ClinicalTrials.gov identifier
CCR2	Plozalizumab	Southwest Oncology Group	II	Metastatic cancer, unspecified adult solid tumor	Completed	NCT01015560
CCL2	Carlumab	Centocor Research & Development, Inc.	II	Prostate cancer	Completed	NCT00992186
CSF-1R	AMG820	Amgen	I	Solid tumors	Completed	NCT0144404
	Emactuzumab (RG7155)	Roche	I	Solid tumors	Completed	NCT01494688
	IMC-SC4	Eli Lilly	I	Breast and prostate cancer	Active	NCT02265536
CD40	SEA-CD40	Seagen Inc.	I	Non-small-cell lung carcinoma, squamous solid tumors	Active	NCT02376699
	LVGN7409	Lyvgen Biopharma Holdings Limited	I	Solid tumors	Active	NCT04635995
	CDX-1140	Celldex Therapeutics	I/II	Melanoma	Active	NCT04364230
	APX005M	Apexigen, Inc.	II	Soft tissue sarcoma	Active	NCT03719430
	ADC-1013	Janssen Research & Development, LLC	I	Advanced solid neoplasms	Active	NCT02829099
	ChiLob 7/4	Cancer Research UK	I	B-cell lymphoma	Completed	NCT01561911
	Selicrelumab	Hoffmann-La Roche	I/II	Pancreatic adenocarcinoma	Active	NCT03193190
FcγRIIb	BI-1206	BioInvent International AB	I/II	Indolent B-cell non-Hodgkin lymphoma	Active	NCT03571568
SIRPα	BI 765063	OSE Immunotherapeutics	I	Solid tumor	Active	NCT03990233
	CC-95251	Celgene	I	Neoplasms	Active	NCT03783403
	GS-0189	Gilead Sciences	I	Non-Hodgkin lymphoma	Active	NCT04502706
VISTA	CI-8993	Curis, Inc.	I	Solid tumor	Active	NCT04475523

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
