# Peer review of "Remodeling the Tumor Myeloid Landscape to Enhance Antitumor Antibody Immunotherapies"

_cancers, 2021, doi:10.3390/cancers13194904_

Round 1
Reviewer 1 Report
In this review the authors discuss the most promising new strategies targeting the cell surface molecules present on tumor associated macrophages (TAM) with agonistic mAb and small molecules to augment tumor targeting mAb immunotherapies and overcome therapeutic resistance. This is a well-written, comprehensive and informative review. I would suggest that the authors can throw some light on the plasticity of macrophages in their review, which is an important factor.
Author Response
We would like to thank Reviewer 1 for their kind comments and suggestions. Indeed, macrophage plasticity is an important aspect of TAM biology and we would have liked to discuss the plasticity of TAM phenotype and function in TME in further detail. However, this is not possible due to the current length of this manuscript and in addition to the focus of this review being primarily on Phagocytosis Checkpoints. We do allude to the transient and dynamic nature of TAM phenotype and activation states at lines 82, 130 and 743.
Best wishes,
Reviewer 2 Report
The authors have made a considerable effort in outlining the various aspects of macriphage therapy for anti cancer therapy and the schematics are very intuitive in giving the viewer a very concise knowledge about the vast area of macrophage therapy.
However they should include the below mentioned references in relation to macrophage cellular therapies for the treatment of cancer.
References to be added
doi:10.1038/s41467-021-20893-2
doi:10.1038/s41587-020-0462-y
Author Response
We would like to thank Reviewer 2 for their kind comments. Thank you for suggesting these very important studies and we have now referenced the exciting work regarding CAR T-Cell mediated depletion of immunosuppressive TAMs to treat tumors, in this manuscript. Best wishes,
Reviewer 3 Report
The topic is complex and I realize that its simplification is very difficult however, the differences in hematologic neoplasms compared to solid tumors and on Cancer direct targeting mAbs such as Rituximab, Herceptin, and Cetuximab compared with immune checkpoint inhibitors, are not clearly highlighted and addressed. The same I would say for different solid tumor types and moreover for different metastatic sites.
So far, there is no evidence that all tumors responding to immune checkpoint inhibitors present polarized M1 TAM and all resistant tumors, M2 TAM. M2 TAM could be the driver of immunotherapy failure in some sites and in some phases of tumor evolution. However, the hard task is to develop a way to capture the TME in its plasticity during tumor progression.
Anyway, this review provides a synthesis of the more relevant literature on this topic, and considering that more than 97,000 papers have been published in this field I congratulate the authors.
Author Response
We would like to thank Reviewer 3 for their kind comments. We agree that the requirements of antitumor TAM functions, and activations states may differ for ‘direct targeting’ versus ‘checkpoint inhibitor’ mAb therapies, and in different tumour niches and types. Therefore, superimpositions of simple dichotomies regarding TAM activation states do not fully address these complexities. Certainly, the manipulation of TAM activation states in experimental models and in patients warrant further study, to address the concerns raised by the reviewer. mAb therapeutics targeting TAMs are still in early stages of development and their performance in early phase trials in combination with direct targeting mAb or immune checkpoint inhibitors will allow the optimal TAM activation states to be ascertained for established mAb therapies in different types of malignant disease. Best wishes,
Reviewer 4 Report
A well written, very comprehensive review on the TME and strategies to enhance the anti-tumor capacity of TAMs through the use of monoclonal antibody therapies. The review is well organized and overall is easy to follow. There are no major concerns. There are some minor changes that once made will enhance the quality of the manuscript. They are as follows:
1) The title has two different fonts, this should be corrected.
2) The Figures and tables are very low resolution. It makes them hard to read. For instance, in figure 2 "blood vessel" is very hard to make out, and most of the text is blurry. The same can be said for figure 3. "ITAM and ITIM" are very hard to see.
3) Line 13 in the abstract, the sentence that begins with "For IgG...". This sentence seems out of place for the abstract, it would read better if removed.
4) There are multiple instances were mAB is used instead of mABs. Make sure to use mABs.
5) Line 55 in the introduction. ADCP is identified as antibody-dependent cellular cytotoxicity. It should be antibody-dependent cellular phagocytosis.
6) Line 84. foetal should be fetal
7) Line 124. Remove the word "the"
8) Line 173. Remove the word "to"
9) Line 197. Should mention that the tumors are xenografts.
10) There are multiple examples of run-on sentences with overuse of commas. Consider breaking up long sentences in to multiple sentences.
11) Some sentences should be re-written, as they are hard to read. 2 examples are: Line 223 "in the context of..." and Line 237 "However, additionally...".
Author Response
We would like to thank Reviewer 3 for their kind comments and a thorough review of the manuscript. We have attempted to correct the typos and issues with the resolution of the figures. Best wishes,
Round 2
Reviewer 1 Report
The authors have responded to the critique
Reviewer 2 Report
The authors have made the recommended changes to
to the manuscript to my satisfaction.
Reviewer 4 Report
The Authors have made the minor changes mentioned in the first report. The Figures look great and are now are very easy to ready